# The Prevalence of Back Pain in Patients Operated on Due to Colorectal Cancer Depending on the Type of Surgical Procedure Performed

**DOI:** 10.3390/cancers15082298

**Published:** 2023-04-14

**Authors:** Iwona Głowacka-Mrotek, Michał Jankowski, Bartosz Skonieczny, Magdalena Tarkowska, Dorota Ratuszek-Sadowska, Anna Lewandowska, Tomasz Nowikiewicz, Karol Ogurkowski, Wojciech Zegarski, Magdalena Mackiewicz-Milewska

**Affiliations:** 1Department of Rehabilitation, Nicolaus Copernicus University in Toruń, Collegium Medicum in Bydgoszcz, 85-094 Bydgoszcz, Poland; 2Department of Surgical Oncology, Nicolaus Copernicus University in Toruń, Collegium Medicum in Bydgoszcz, 85-094 Bydgoszcz, Polandskonieczny@gumed.edu.pl (B.S.);; 3Department of Urology, Nicolaus Copernicus University in Toruń, Collegium Medicum in Bydgoszcz, 85-094 Bydgoszcz, Poland; m.sowa@cm.umk.pl

**Keywords:** low back pain, surgery, colorectal cancers, open surgery, laparoscopy

## Abstract

**Simple Summary:**

Our study aimed to assess back pain in patients undergoing surgery for colorectal cancer following anterior resection of rectum (AR), laparoscopic anterior resection of rectum (LAR), Hartmann’s procedure (HART), or abdominoperineal resection of rectum (APR). The studied patients were examined three times: prior to surgery (I), six months after surgery (II), and one year after surgery (III). The analysis of study results revealed that an increase in the degree of disability and functioning impairment occurred in all groups between time points I and II. A reduction in low back pain was observed one year after the procedure in the LAR group.

**Abstract:**

Purpose: Low back pain presents a serious challenge for numerous medical specialties. The purpose of this study was to assess disability due to low back pain in patients operated on due to colorectal cancer depending on the type of surgery performed. Methods: This prospective observational study was carried out in the period of July 2019 through March 2020. Included in the study were patients with colorectal cancer for scheduled surgeries including anterior resection of rectum (AR), laparoscopic anterior resection of rectum (LAR), Hartmann’s procedure (HART), or abdominoperineal resection of rectum (APR). The Oswestry Low Back Pain Disability Questionnaire was used as the research tool. The study patients were surveyed at three time points: before surgery, six months after surgery, and one year after surgery. Results: The analysis of study results revealed that an increase in the degree of disability and functioning impairment occurred in all groups between time points I and II, with the differences being statistically significant (*p* < 0.05). The inter-group comparative analysis of the total Oswestry questionnaire scores revealed statistically significant differences, with the impairment of function being most severe within the APR group and least severe within the LAR group. Conclusion: The study results showed that low back pain contributes to impaired functioning of patients operated on due to colorectal cancer regardless of the type of procedure performed. A reduction in the degree of disability due to low back pain was observed one year after the procedure in patients having undergone LAR.

## 1. Introduction 

Each year, about 18,000 new cases of colorectal cancer are diagnosed in Poland. The radical treatment of this cancer is based on surgical resection of the tumor. Depending on the tumor location and the stage of the disease, the surgical treatment may be supplemented by neoadjuvant or adjuvant radio- or chemotherapy [1].

Colorectal cancer resection procedures may be carried out by means of classic (open) or laparoscopic methods. As supported by numerous publications, the laparoscopic approach is believed to be associated with a number of advantages, namely reduced intraoperative blood loss, reduced pain in the postoperative period, and faster mobilization of patients after surgery [1,2,3]. 

Regardless of the type of surgical procedure, tissue continuity is interrupted in all patients. The damage consists of the incision of the skin, subcutaneous tissue, fascia, and abdominal muscles, with the difference between the laparoscopic and the classic (open) method consisting merely the extent of the injury. Muscle mass loss is also observed in patients following the procedure [4]. The abdominal cavity is built of consecutive layers of skin, subcutaneous tissue, transverse muscles, abdominal oblique muscles, and the fascia. The role of the abdominal muscles consists primarily in supporting the lateral, anterior, and posterior abdominal cavity walls as well as intraabdominal organs. Abdominal muscles also support the abdominal prelum, contribute to breathing, and stabilize the spine within the lumbosacral region [5]. The extensive abdominal muscle corset determines the mobility of the trunk, i.e., the ability to twist and bend one’s body, and contributes to maintaining proper body posture [6]. The postoperative period involves immobilization and bed confinement and is associated with reduced protein and energy supply as well as fatigue. The postoperative limitation of normal activity may last for months and have a negative impact on the quality of life and functioning of patients [7].

One should also note that any type of abdominal surgery is associated with creating incision(s) leading to subsequent scar formation. Surgical scars and impairment of abdominal muscles may contribute to lumbar pain being experienced by patients operated on due to colorectal cancer. Lumbar back pain is a serious problem leading to serious costs to the healthcare systems, workplace absenteeism as well as disability in the long run [8].

## 2. Objective of the Study

The objective of this study was to assess back pain in patients operated on due to colorectal cancer depending on the type of procedure performed including anterior resection of rectum (AR), laparoscopic anterior resection of rectum (LAR), Hartmann’s procedure (HART), or abdominoperineal resection of rectum (APR). According to the authors’ knowledge, no publications are available on the subject of the prevalence of low back pains depending on the type of surgery. 

## 3. Material and Methods

### 3.1. Study Design

This prospective study was carried out subject to approval issued by the Bioethics Committee at the Nicolaus Copernicus University in Toruń (Decision No. 283/2019) and in accordance with the principles of good clinical practice and ethical considerations of the Helsinki Declaration. 

### 3.2. Study Enrollment

The recruitment lasted from July 2019 through March 2020. Included in the study were patients with colorectal cancer admitted to the Clinical Department of Oncological Surgery of the Oncology Centre in Bydgoszcz for scheduled surgeries including anterior resection of the rectum, laparoscopic anterior resection of the rectum, Hartmann’s procedure, or abdominoperineal resection of the rectum. The detailed outline of the study recruitment process is presented in Figure 1.

The inclusion criteria were met by 163 patients. A total of 58 patients were lost to follow-up or refused further participation after the first time point while another 10 patients were lost to follow-up or refused further participation after the first time point. After the first stage of the study, 25 patients did not answer the phone or did not consent to further participation in the study. Finally, a total of 96 patients were included in the analysis.

The inclusion criteria were as follows:-consent to participate in the study;-age over 18 years;-colorectal cancer patients qualified for scheduled surgeries including anterior resection of rectum, laparoscopic anterior resection of the rectum, Hartmann’s procedure, or abdominoperineal resection of the rectum;-the primary character of colorectal cancer surgery;-good mobility, Zubrod-ECOG-WHO performance score of 0–1;-colorectal cancer of stage I–III as per preoperative assessment.

The exclusion criteria were as follows:-disseminated disease (stage IV cancer);-ASA score of 4 or higher;-intraoperative conversion from laparoscopic to open surgery;-severe cardiovascular, pulmonary, orthopedic, and neurological pathologies;-cognitive impairment;-local recurrence or distant metastases in the follow-up period.

Patients meeting the inclusion criteria were given the opportunity to participate in the study. Having talked to the investigators and having read the patient information sheet, the participants signed the written informed consent forms. Next, the patients were asked to complete the Oswestry Disability Index questionnaire, also known as the Oswestry Low Back Pain Disability Questionnaire. The questionnaire contained 10 questions on pain intensity, personal care, lifting, walking, sitting, standing, sleeping, social life, travel, and pain intensity changes. Each question was accompanied by 6 optional answers scored 0 to 5 with higher scores corresponding to higher degrees of disability. The sum of the answers to the 10 questions amounted to a minimum of 0 points and a maximum of 50 (percentage range 0 to 100%). The disability levels were defined as follows:
0–20%—minimal disability;21–40%—moderate disability; 41–60%—severe disability;61–80%—crippled;81–100%—complete motor impairment.

Clinical and sociodemographic data as available in the patients’ records, such as age, weight, height, BMI, hospitalization time, gender, pre- and post-operative treatment, postoperative complications, and disease staging, were also used in the analysis.

The patients were surveyed at three time points: the first questionnaire was completed in the hospital prior to surgery while the second and third questionnaires were completed six months and one year after the procedure, respectively. The responses at the latter two time points were collected by means of the CATI- Computer Assisted Telephone The patients were interviewed by telephone due to the ongoing coronavirus pandemic. Patients, by signing informed consent, knew at what point in the procedure telephone contact would occur. The persons interviewing the patients were the 4 authors of the manuscript.

ERAS protocol was extensively followed at the study site. Prior to surgery, the attending physician recommended that the patients receive preoperative protein supplementation, physical activity was maintained before and after the procedure in the recovery, a carbohydrate-rich drink was administered up to 2 h prior to anesthesia, preoperative fasting was avoided, and timely restitution of oral nutrition following surgery was recommended. Attention was paid to reducing the dwelling times of abdominal drains and urinary catheters, avoiding routine use of gastric probes, and discharging hospitalized patients in a timely manner. Recommendations for home-based nutritional management were given out to patients upon discharge.

### 3.3. Statistical Methods

Statistical analyses were carried out using the PQStat statistical package version 1.8.2.232. 

Inter-group differences in age, weight, height, and BMI were analyzed by means of the unifactorial analysis of variance and the post-hoc Tukey test whereas differences in hospitalization times were analyzed by means of the Kruskal–Wallis test with the post-hoc Dunn–Bonferroni’s test.

For qualitative variables, such as gender, pTNM, pre- and post-operative management, complications, and disease staging, the chi-squared test or the exact Fisher’s test was used.

The Oswestry questionnaire scores were compared between patient groups using the Kruskal–Wallis test while the differences between individual time points were analyzed using the Friedman test. The Dunn–Bonferroni test was used for post-hoc analysis in both cases.

The disability scales (%) between individual surgery groups were compared using the Kruskal–Wallis test. Pre- and post-operative scores were compared using the Friedman test.

A test probability of *p* < 0.05 was defined as statistically significant whereas a test probability of *p* < 0.01 was defined as highly significant.

## 4. Results

A total of 96 patients were included in the analysis, with 37 patients operated on by means of anterior resection of the rectum, 24 patients operated on by means of laparoscopic anterior resection of rectum, 13 patients operated on by means of Hartmann’s procedure, and 22 patients operated on by means of abdominoperineal resection of the rectum.

The study groups did not differ in age, weight, height, BMI, or postoperative hospitalization times (*p* > 0.05). The characteristics of the study groups are presented in Table 1.

The study groups were analyzed in terms of qualitative data. No statistically significant differences were observed between the study groups regarding the patients’ gender, type of postoperative management, and disease staging (*p* > 0.05) whereas statistically significant differences were observed with regard to the type of preoperative management (*p* = 0.0072). The results are presented in Table 2.

Table 3 shows answers to the individual questions within the Oswestry questionnaire as provided in different study groups and at different study time points. Highly statistically significant differences were observed at different time points within each group with regard to pain intensity, personal care, lifting, walking, and sitting (*p* < 0.01); highly statistically significant differences were also observed at different time points with regard to standing within the LAR group (*p* = 0.0002), sleeping within the APR group, pain intensity changes in the APR, AR, and LAR groups (*p* < 0.01), travel in the APR group, and social life in the APR, AR, and LAR groups. Statistically significant differences were observed with regard to sleeping in the AR group.

Presented in Table 3 are the overall results of the Oswestry questionnaire. No significant differences were observed between study groups at the first and the second time points. At the third time point, highly statistically significant differences were observed between the study groups. The highest scores translating to the highest levels of pain were recorded in the APR group. The lowest scores corresponding to the best functioning were reported in the LAR group. 

Listed below the median values are the letter codes indicating the homogeneity of groups as per the Dunn–Bonferroni test results. The first row below the median values provides the results of the Kruskal–Wallis test whereas the second row below the median values provides the results of the Friedman test.

At the next stage of the analysis, patients within the study groups (APR, AR, LAR, and HART) were assigned to appropriate disability levels. No statistically significant differences in the disability level were observed at the first and the second time points between groups operated on using different surgical approaches. The results are shown in Figure 2 and Figure 3. 

Statistically significant differences were observed with regard to the disability levels depending on the type of surgery (*p* < 0.01). The lowest scores translating to the lowest disability level were observed within the LAR group. The results are presented in Figure 4.

The next stage of the analysis consisted in analyzing the differences in disability levels depending on the study time point. A highly significant change (*p* < 0.01), i.e., an increase in the disability level between time points I and II was observed in the group having undergone the APR. A highly significant change (*p* < 0.01), i.e., an increase in the disability level between time points I and II was observed in the AR group. Highly significant changes (*p* < 0.01), i.e., an increase in the disability level between time points I and II and a reduction in the disability level between time points II and III were observed in the LAR group. No significant differences (*p* < 0.05) were observed between individual time points in the HART group. 

## 5. Discussion

The study results showed that low back pain is observed in patients operated on due to colorectal cancer regardless of the type of procedure performed. The greatest intensity of pain was observed six months after the surgical procedure. A reduction in low back pain was observed one year after the procedure in the LAR group. The highest levels of pain in the lumbar area were observed in patients having undergone the APR procedure. 

According to our knowledge, this was the first study to evaluate the intensity of low back pain in patients operated on due to colorectal cancer depending on the type of surgery performed. Patient evaluation was carried out within a yearly time frame. The Oswestry questionnaire which had been successfully used in numerous previous studies [9,10,11], was used as the study tool.

The study showed that the intensity of pain was highest within the APR group. This was obviously due to the fact that APR was the most extensive of all analyzed procedures, involving a 15–20-cm incision being made within the medial line, sigmoidostomy being formed, and an additional perineal incision, 10–15 cm in length, being made to remove the sphincter apparatus [12].

The bowel movement pattern is also changed in ostomy patients. The lack of tenesmus upon defecation results in the inactivation of the muscles of the pelvic diaphragm, the abdomen, and the respiratory diaphragm. The dyssynergy of these muscle groups may lead to disturbed postural control and complications resulting from abnormal intraabdominal pressures; these, in turn, may either trigger or contribute to the exacerbation of lower back pain. Unfortunately, no studies on the presented subject could be found in the available literature. 

Other authors reported on cases of back pain in patients with ostomies [13,14]. In our study, ostomies were formed in patients having undergone Hartmann’s procedure or abdominoperineal resection surgeries. No differences were observed with regard to the disability levels within the HART group, as well as no differences were observed within this group with regard to the Oswestry questionnaire subscales of standing, sleeping, social life, travel, and changes in the intensity of pain as well as to the total Oswestry scores. However, the group of patients having undergone Hartmann’s procedure was less numerous than the other groups, which makes it difficult to interpret the results or draw any firm conclusions on the basis thereof.

An increase in back pain was observed in our study between time points I and II in the APR, AR, and LAR groups. The back pain might have been due to the disrupted continuity of muscles which affects all patients operated on due to colorectal cancer. The continuity of muscles is disrupted permanently in ostomy patients. This may result in the loss of motor efficiency and impairment of the motor apparatus. Metabolic stress triggered by the surgical procedure itself and the preparation thereto may lead to body weight loss, progressive fatigue, and respiratory impairment [15,16]. Muscle weakness, muscle tissue loss, and weight loss are among other factors contributing to the prevalence of back pain [17,18]. Prado et al. demonstrated that the loss of muscle mass may be related to poor outcomes [19].

Another factor responsible for pain in patients having undergone surgical procedures consists of changes in intraabdominal pressure. Intraabdominal pressure, if maintained at appropriate levels during rest and physical activity by an appropriate tone of the abdominal muscles and the pelvic and respiratory diaphragm, warrants appropriate stability of the spine. Disruption of the abdominal walls leads to changes in intraabdominal pressure, potentially translating to the development of low back pain [20,21]. In ostomy patients, the abdominal cavity is permanently open. 

Scars are also formed at surgery sites, their nature depending on the surgical procedure as well as on the type of the healing process [22,23]. The type of procedure may also influence the size and the number of scars. Patients undergoing abdominoperineal resection of the rectum have incisions made within the medial line and the perineal region; an ostomy is also formed as part of the procedure [9]. This might explain the most severe intensity of lower back pain in this study group. The structure of the scar tissue and the extent of the scar may give rise to local pain as well as pain being transferred to different areas of the body [24]. This may be due to structural distortions and mobility restrictions within the scars and the surrounding tissues. The lack of tissue and fascial mobility combined with the disturbed mobility of the nervous system cells penetrating through all body layers may lead to the disruption of global postural patterns. Numerous studies suggest that manual treatment of scars by means of physiotherapy or invasive techniques such as surgical procedures may help patients regain normal pain-free functioning [25,26,27]. However, patients operated on due to colorectal cancer are not routinely referred to physiotherapeutic management of surgical incision scars. No scar physiotherapy had been offered to patients in our study. Back pain can also be explained by the patients’ activity, body weight, and muscle mass being reduced after surgery. These problems are more common in patients operated on using classic methods.

As shown by our study, surgical treatment is responsible for the development of back pain regardless of the surgical approach (AR vs. LAR). However, the results observed in the LAR group were better than in the remaining groups. The benefits of laparoscopic surgery had been presented in earlier studies by other authors [1,2,3,25]. The better results observed in the laparoscopic anterior resection of the rectum group can be explained by the lower rate of post-operative pulmonary and circulatory complications leading to more timely activation of patients [28,29]. Another factor influencing the prevalence of pain in patients operated on due to colorectal cancer consists in the lack of daily physical activity which is similar to that observed in the geriatric population. The average age of the study patients was slightly higher than 63 years and was typical for colorectal cancer patients [30]. The importance of reduced activity, especially within elderly patients, was also highlighted by Ekblom-Bak et al., whose study of compliance with physical activity recommendations within an elderly population showed that minimum physical activity requirements were met by as few as 7% of subjects [31]. 

In addition to postural muscle weakness, chronic postoperative pain associated with perioperative nerve transection can be also a cause of spinal pain. Chronic postoperative pain is a significant clinical problem. According to Lois et al., a factor leading to postoperative spinal pain in patients treated for colorectal cancer is the presence of pain in the preoperative period [32]. Similar findings were observed by Gerbershagen et al. The study involved patients after prostatectomy surgery [33]. Other authors also cite younger age, female gender, and psychological and genetic factors as reasons for chronic postoperative pain [34,35].

Observations were carried out over a period of one year after surgery and should be treated as early results of the surgical treatment. Notably, the recurrence of colorectal cancer is most frequently observed within the first 2–3 years after surgical treatment, with early recurrences being uncommon [36]. On the other hand, pain, including low back pain, may be associated with local recurrence or spread of the disease [37].

In our study, due to the coronavirus pandemic, contact with patients in the second and third phases of the study was made using the CATI method. To ensure that there was no bias during the study, the callers of the patients had no data on the procedures provided to them. The CATI method was considered by the authors to be a safe method to survey patients during a pandemic. Other studies have previously highlighted the usefulness of this method [38].

## 6. Study Limitations

The study was burdened by certain limitations, the greatest consisting of the small number of the study group. Another limitation is the loss of contact with many patients between the first and second stages of the study. However, it should be noted that the study group consisted of consecutive patients operated on in a single high-volume center, and thus errors due to improper, non-standardized surgical procedures could be avoided. Certainly, a multi-center study could verify the collected data.

Another limitation consists of the fact that no patients operated on by means of laparoscopic abdominoperineal resection of the rectum were included in the study group. This was due to the fact that the annual number of procedures of this type within the period of the study was lower than 15. Additionally, the number of study patients operated on by means of Hartmann’s procedure was small.

## 7. Conclusions

This study showed that lower back pain is a problem in patients having undergone surgeries due to colorectal cancer. The most affected group of patients consisted of those who had undergone abdominoperineal resection of the rectum. Laparoscopic surgery led to the pain symptoms being reduced as compared with other patient groups one year after surgery.

## Figures and Tables

**Figure 1 cancers-15-02298-f001:**
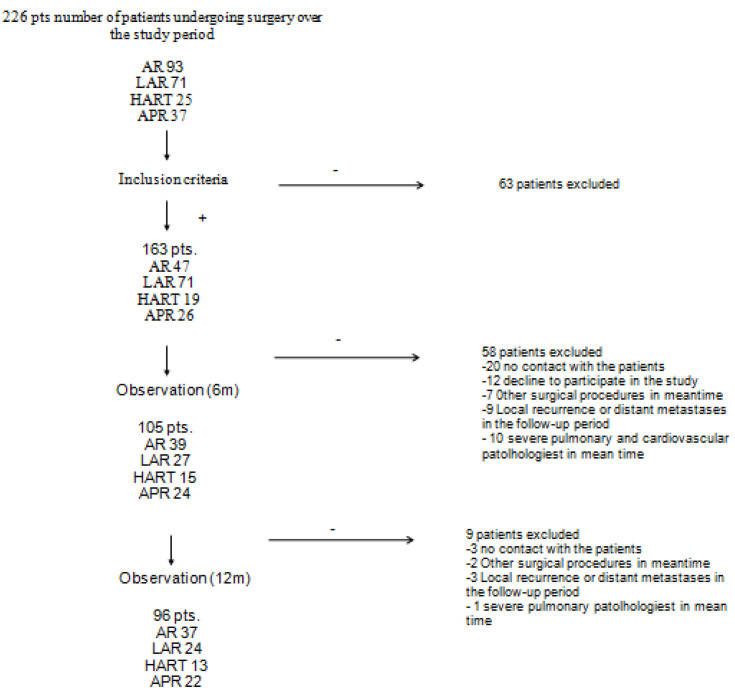
Study flow chart. APR—abdominoperineal resection; LAR—laparoscopic anterior resection; AR—anterior resection; HART—Hartmann Procedure.

**Figure 2 cancers-15-02298-f002:**
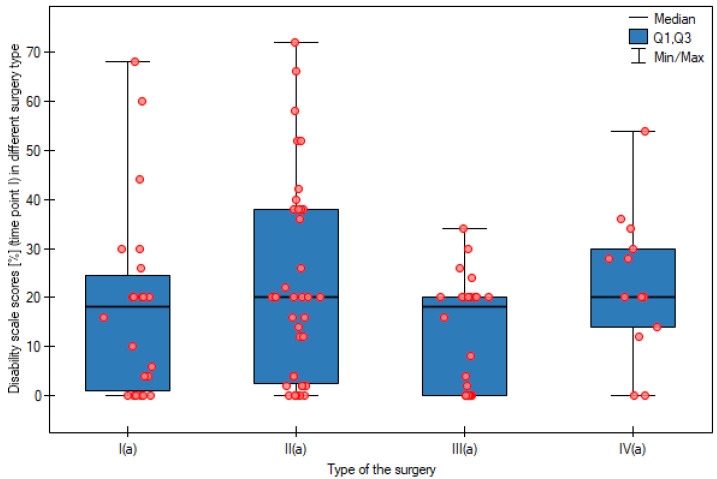
Disability scale scores (%) (time point I) in different surgery type groups. I(a)—APR—abdominoperineal resection of the rectum; II(a)—AR—anterior resection of the rectum; III(a)—LAR—laparoscopic anterior resection of the rectum; IV(a)—HART—Hartmann’s procedure; *p*—statistical significance level.

**Figure 3 cancers-15-02298-f003:**
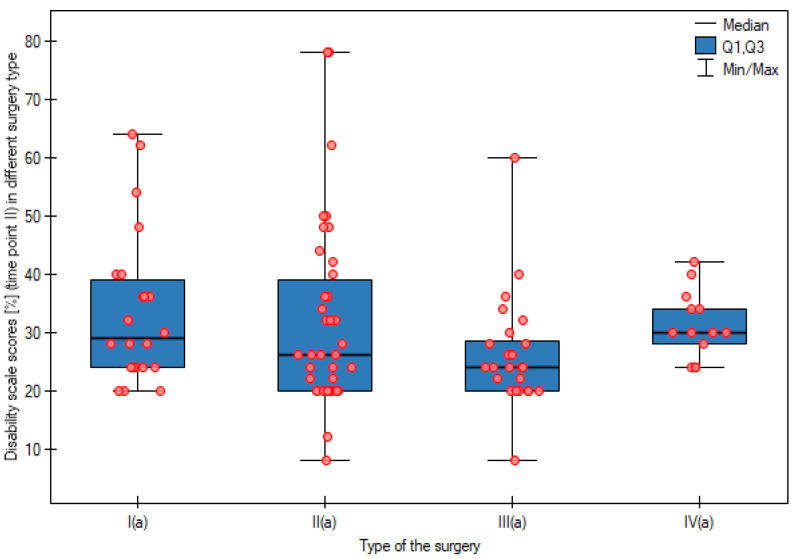
Disability scale scores (%) (time point II) in different surgery type groups. I(a)—APR—abdominoperineal resection of the rectum; II(a)—AR—anterior resection of the rectum; III(a)—LAR—laparoscopic anterior resection of the rectum; IV(a)—HART—Hartmann’s procedure; *p*—statistical significance level.

**Figure 4 cancers-15-02298-f004:**
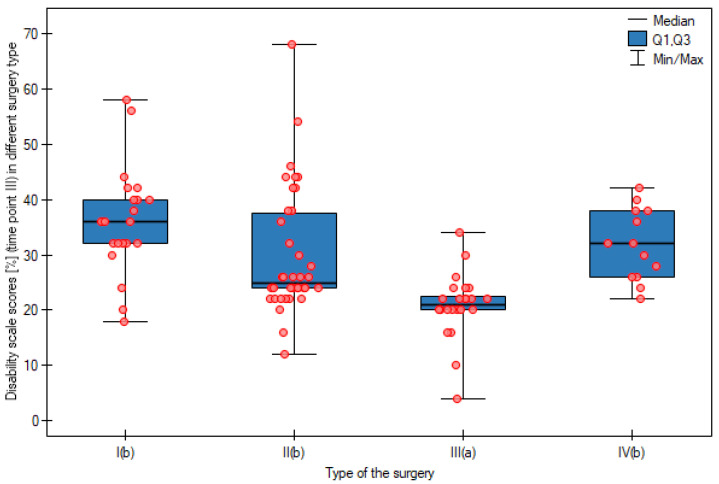
Disability scale scores (%) (time point III) in different surgery type groups. I(b)—APR—abdominoperineal resection of the rectum; II(b)—AR—anterior resection of the rectum; III (a)—LAR—laparoscopic anterior resection of the rectum; IV(b)—HART—Hartmann’s procedure; *p*—statistical significance level.

**Table 1 cancers-15-02298-t001:** Quantitative clinical data and inter-group differences in individual patient groups.

Scale	Group	Mean	S.D.	ANOVA **
Age	APR	63.2	7.7	*p* = 0.3781
	AR	64.5	6.6	
	LAR	63.9	11.4	
	HART	68.1	5.9	
Weight	APR	76.4	14.8	*p* = 0.2340
	AR	83.2	16.2	
	LAR	76.2	14.3	
	HART	76.6	17.0	
Height	APR	170.6	8.8	*p* = 0.6031
	AR	170.3	10.5	
	LAR	168.4	10.3	
	HART	166.5	11.1	
BMI	APR	26.4	6.0	*p* = 0.2748
	AR	28.6	4.7	
	LAR	26.7	3.8	
	HART	27.3	3.7	
Postoperative hospitalization time **	APR	10.0	6.0	*p* = 0.1020
	AR	9.2	6.3	
	LAR	7.4	4.2	
	HART	9.0	4.9	

APR—abdominoperineal resection of the rectum; AR—anterior resection of the rectum; LAR—laparoscopic anterior resection of the rectum; HART—Hartmann’s procedure; *p*—statistical significance level. ** in the case of the “hospitalization time” variable, the analysis involved the Kruskal–Wallis test with the Dunn–Bonferroni post-hoc test.

**Table 2 cancers-15-02298-t002:** Qualitative clinical data and inter-group differences in individual patient groups (abdominoperineal resection of the rectum, anterior resection of rectum, laparoscopic anterior resection of the rectum, and Hartmann’s procedure).

		Type of Surgical Procedure								^1^ Chi-Squared Test ^2^ Exact Fisher’s Test
		APR		AR		LAR		HART		
		N	%	N	%	N	%	N	%	
Sex	Female	7	31.8	11	29.7	9	37.5	3	23.0	^1^*p* = 0.8185
	Male	15	68.2	26	70.3	15	62.5	10	77.0	
Type of preoperative treatment	none	1	4.5	10	27.0	9	37.5	0	0	^2^*p* = 0.0072
	chemotherapy	0	0	0	0	0	0	1	7.7	
	radiochemotherapy	13	59.1	19	51.3	6	25	6	46.1	
	radiotherapy	8	36.4	8	21.6	9	37.5	6	46.1	
Type of postoperative treatment	none	16	72.7	25	67.6	15	65.2	9	69.2	^2^*p* = 0.6871
	chemotherapy	6	27.3	12	32.4	7	30.4	3	23.1	
	radiochemotherapy	0	0	0	0	1	4.3	1	7.7	
	radiotherapy	0	0	0	0	0	0	0	0	
Cancer stagingp TNM/ypTNM	0	0	0	4	10.8	1	4.2	1	7.7	^2^*p* = 0.2308
	I	11	52.4	7	18.9	9	37.5	4	30.8	
	IIA	3	14.3	14	37.8	4	16.7	5	38.5	
	IIB	1	4.8	0	0	0	0	0	0	
	IIIA	1	4.8	0	0	0	0	0	0	
	IIIB	3	14.3	6	16.2	7	29.2	2	15.4	
	IIIC	2	9.5	6	16.2	3	12.5	1	7.7	

APR—abdominoperineal resection of the rectum; AR—anterior resection of the rectum; LAR—laparoscopic anterior resection of the rectum; HART—Hartmann’s procedure; *p*—statistical significance level.

**Table 3 cancers-15-02298-t003:** Results in the individual subscales of the Oswestry questionnaire in individual groups (abdominoperineal resection of the rectum, anterior resection of rectum, laparoscopic anterior resection of rectum, and Hartmann’s procedure) and at different study time points.

	Group	Term						Friedman Test
		I Examination		II Examination		III Examination		
		Arithmetic Mean	Median	Arithmetic Mean	Median	Arithmetic Mean	Median	
Pain intensity	APR	0.86	1.00 a a	2.09	2.00 b b	3.05	3.00 c c	*p* < 0.0001
	AR	1.49	1.00 a a	1.89	2.00ab ab	2.27	2.00 b b	*p* < 0.0001
	LAR	0.75	1.00 a a	1.42	1.00 a ab	1.50	1.00 a b	*p* = 0.0001
	HART	1.62	1.00 a a	2.08	2.00ab ab	2.62	2.00 bc b	*p* = 0.0080
Kruskal–Wallis test		*p* = 0.0411		*p* = 0.0413		*p* < 0.0001		-
Personal care	APR	0.73	1.00 a a	1.23	1.00 a a	1.82	2.00 a b	*p* < 0.0001
	AR	0.89	1.00 a a	1.32	1.00 a b	1.54	1.00 a b	*p* < 0.0001
	LAR	0.54	0.50 a a	1.04	1.00 a ab	1.29	1.00 a b	*p* < 0.0001
	HART	0.77	1.00aa	1.38	1.00aab	1.85	2.00ab	*p* = 0.0024
Kruskal–Wallis test		*p* = 0.6899		*p* = 0.2232		*p* = 0.0277		-
Lifting	APR	0.95	1.00 a a	1.95	2.00 a b	3.18	3.00 b b	*p* < 0.0001
	AR	1.32	1.00 a a	2.00	2.00 a b	2.30	2.00 b b	*p* < 0.0001
	LAR	0.58	1.00 a a	1.58	1.00 a b	1.33	1.00 a b	*p* < 0.0001
	HART	1.31	1.00 a a	2.62	2.00 a ab	3.00	3.00 b b	*p* = 0.0013
Kruskal–Wallis test		*p* = 0.1782		*p* = 0.1174		*p* = 0.0001		-
Walking	APR	0.64	0.50 a a	1.68	1.00 a b	1.45	1.00 b b	*p* < 0.0001
	AR	0.86	1.00 a a	1.57	1.00 a b	1.22	1.00 ab ab	*p* < 0.0001
	LAR	0.54	1.00 a a	1.50	1.00 a b	0.92	1.00 a a	*p* < 0.0001
	HART	0.69	1.00 a a	1.31	1.00 a a	1.31	1.00 ab a	*p* = 0.0043
Kruskal–Wallis test		*p* = 0.4671		*p* = 0.7906		*p* = 0.0254		-
Sitting	APR	1.27	1.00 a ab	2.36	2.00 a b	0.95	1.00 a a	*p* = 0.0020
	AR	1.19	1.00 a a	1.89	2.00 a b	1.22	1.00 a ab	*p* = 0.0001
	LAR	0.75	1.00 a a	1.67	2.00 a b	0.92	1.00 a ab	*p* = 0.0001
	HART	1.31	1.00 a a	2.15	2.00 a a	1.23	1.00 a a	*p* = 0.0043
Kruskal–Wallis test		*p* = 0.5476		*p* = 0.5076		*p* = 0.3011		-
Standing	APR	1.27	1.00 a a	1.55	1.00 a a	1.32	1.00 b a	*p* = 0.1381
	AR	1.38	1.00 a a	1.59	1.00 a a	1.32	1.00 ab a	*p* = 0.0529
	LAR	0.75	1.00 a a	1.50	1.00 a b	0.83	1.00 a ab	*p* = 0.0002
	HART	1.38	1.00 a a	1.46	1.00 a a	1.15	1.00 ab a	*p* = 0.5866
Kruskal–Wallis test		*p* = 0.3333		*p* = 0.9917		*p* = 0.0266		-
Sleeping	APR	0.73	1.00 a a	1.55	1.00 a b	1.18	1.00 a ab	*p* = 0.0003
	AR	1.14	1.00 a a	1.51	1.00 a a	1.38	1.00 a a	*p* = 0.0142
	LAR	0.63	1.00 a a	1.13	1.00 a a	0.92	1.00 a a	*p* = 0.0039
	HART	1.15	1.00 a a	1.15	1.00 a a	1.31	1.00 a a	*p* = 0.3114
Kruskal–Wallis test		*p* = 0.2426		*p* = 0.1754		*p* = 0.0925		-
Social life	APR	0.50	0.00 a a	1.55	1.00 a b	1.50	1.00 b b	*p* < 0.0001
	AR	0.89	1.00 a a	1.32	1.00 a a	1.19	1.00 ab a	*p* = 0.0082
	LAR	0.63	1.00 a a	1.04	1.00 a a	1.00	1.00 a a	*p* = 0.0024
	HART	1.00	1.00 a a	1.15	1.00 a a	1.08	1.00 ab a	*p* = 0.3679
Kruskal–Wallis test		*p* = 0.4439		*p* = 0.1965		*p* = 0.0336		-
Traveling	APR	1.05	1.00 a a	1.50	1.00 a ab	1.86	2.00 b b	*p* = 0.0030
	AR	1.14	1.00 a a	1.38	1.00 a a	1.30	1.00 a a	*p* = 0.2040
	LAR	0.67	1.00 a a	1.13	1.00 a a	0.92	1.00 a a	*p* = 0.0305
	HART	1.23	1.00 a a	1.00	1.00 a a	1.00	1.00 a a	*p* = 0.2765
Kruskal–Wallis test		*p* = 0.1709		*p* = 0.1604		*p* < 0.0001		-
Pain intensity changes	APR	1.05	1.00 a a	1.41	1.00 a a	1.77	1.50 b a	*p* = 0.0097
	AR	1.11	1.00 a a	1.59	1.00 a b	1.38	1.00 b ab	*p* = 0.0011
	LAR	0.50	0.00 a a	1.17	1.00 a b	0.79	1.00 a ab	*p* = 0.0006
	HART	0.92	1.00 a a	1.31	1.00 a a	1.38	1.00 b a	*p* = 0.1561
Kruskal–Wallis test		*p* = 0.0738		*p* = 0.2768		*p* = 0.0003		-
Overall results of the Oswestry questionnaire	APR	9.05	9.00 a a	16.86	14.50 a b	18.09	18.00 b b	*p* < 0.0001
	AR	11.41	10.00 a a	16.08	13.00 a b	15.05	13.00 b b	*p* = 0.0002
	LAR	6.33	9.00 a a	13.17	12.00 a b	10.42	10.50 a a	*p* < 0.0001
	HART	11.38	10.00 a a	15.62	15.00 a a	15.92	16.00 b a	*p* = 0.0518
Kruskal–Wallis test		*p* = 0.1712		*p* = 0.0999		*p* < 0.0001		-

APR—abdominoperineal resection of the rectum; AR—anterior resection of the rectum; LAR—laparoscopic anterior resection of the rectum; HART–Hartmann’s procedure; *p*—statistical significance level. a, b—below the median values, there are letter codes indicating groups based on the Friedman test. The same letter for two groups means that they are not significantly different, and if two groups do not share a letter, it means that they are significantly different.

## Data Availability

The datasets generated during and/or analyzed Turing the current study are available from the corresponding author on responsiblereques.

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
