# Peer review of "The Prevalence of Back Pain in Patients Operated on Due to Colorectal Cancer Depending on the Type of Surgical Procedure Performed"

_cancers, 2023, doi:10.3390/cancers15082298_

Round 1
Reviewer 1 Report
This is an interesting study that investigated the prevalence of postoperative lower back pain after rectal cancer surgery. The authors found that all kinds of procedure had increased the disability and functioning impairment over time, and the severity had sustained 1 year after surgery except laparoscopic anterior resection.
Major points:
1. The greatest concern is that the association of functional impairment of abdominal muscles with operative scars and chronic postoperative lumber pain. Did they really contribute to the observed lumbar pack pain after colorectal surgery?? The authors have discussed this in the introduction and discussion: however, this hypothesis may not have been firmly supported by the literature. Functional loss of the abdominal muscles can be directly assessed by muscle strength testing, and anatomical loss of the muscles can also be assessed by measuring muscle area on CT (like sarcopenia). In addition, Lois et al. have suggested that the presence of preoperative pain has significantly impacted on chronic post-surgical pain after colon surgery (Acta Anaesthesiol Scand 2019;63:931-938). The authors are advised to discuss more by collecting supportive evidences.
2. One of the major limitations of this study is the small number of patients included in the analysis. A total of 58 patients were lost to follow-up during 6 months after surgery, and this number was over one-third of the included patients. This fact may lead to questioning the quality of postoperative care in the study protocol.
3. The patients who developed local recurrence and/or distant metastasis in the follow-up period were excluded. It is assumed that the cancer recurrence may not always be associated with chronic or persistent lower back pain. In addition, the patients who underwent converted procedure were also excluded. These individuals may have possibly included in the open group. The authors are advised to comment on #2 and #3 for improved study designing in the future.
Minor points:
1. There are number of misspelling (in English) in the tables and figures, such as “chemio” and “radiochemiotherapy” in Table 2, “Srednia”, “Mediana”, and “Grupa III” in Table 3, and “Mediana” in Figure 2-4.
2. There are inconsistencies in abbreviations of “LAR” and “HART”. These words are supposed to indicate “laparoscopic anterior resection of the rectum” and “Hartmann’s procedure”: however, these abbreviations have appeared in the tables as “ARL” and “HP”, respectively.
3. There are some unclear letters in the tables. What is “Turkey Te” in Table 1? What are the meaning of “a”, “b”, and “ab” in Table 2 and 3?
4. “CATI methods” need to be spelled out when firstly appeared in the manuscript.
5. Figure 1 showed that a total of 95 patients were finally analyzed, but the main text described that 96 patients were included.
Author Response
Dear Reviewer #1,
Thank you for reviewing our article titled ‘The prevalence of back pain in patients operated on due to colorectal cancer depending on the type of surgical procedure performed ‘ We deeply appreciate your opinion as well as constructive comments that contributed to more profound consideration of issues addressed in our publication. The comments in your review will guide us in our future work.
In response to those commentaries we have clarified the following points:
Major points
- Thank you for this comment. In the discussion section we discussed how the presence of preoperative pain affects chronic postoperative pain.
- We agree that we lost a large group of patients between the first and second stages of the study, we have highlighted this in the study limitation. We have expanded Figure 1 reasons for loss of contact with patients.
- We agree that patients undergoing conversion surgery could have been included in the open patient group.
Minor points:
- There are number of misspelling (in English) in the tables and figures, such as “chemio” and “radiochemiotherapy” in Table 2, “Srednia”, “Mediana”, and “Grupa III” in Table 3, and “Mediana” in Figure 2-4.
We have corrected it
- There are inconsistencies in abbreviations of “LAR” and “HART”. These words are supposed to indicate “laparoscopic anterior resection of the rectum” and “Hartmann’s procedure”: however, these abbreviations have appeared in the tables as “ARL” and “HP”, respectively.
We have corrected it.
- There are some unclear letters in the tables. What is “Turkey Te” in Table 1? What are the meaning of “a”, “b”, and “ab” in Table 2 and 3?
We have corrected Table 1 with the ANOVA test.
We have explained the meaning of letter a,b in the legend for tables 2 and 3.
- “CATI methods” need to be spelled out when firstly appeared in the manuscript.
We spelled out the meaning of the CATI method when it first appeared in the manuscript.
- Figure 1 showed that a total of 95 patients were finally analyzed, but the main text described that 96 patients were included.
We have corrected the number of patient in figure 1. We have extended the figure 1 with the additional data
Thanks to the comments received, we were able to refine the publication in terms of the content. All errors mentioned in the review were corrected in the final version of the publication.
Kind regards,
the Authors
Reviewer 2 Report
This is an interesting paper because there have been no attention paid to lower back pain in these group of patients.
Doing research during the pandemic was difficult. However, from a scientific approach it must be discussed how the telephone interviews have influenced the results. Who was making the interviews? If one of the authors bias can have influenced the result of the interview. Were the interviews scheduled so the participant knew when they were called or was it unplanned. This is not discussed at all and it would make it a better paper if this was included in the discussion as well as in the method and material section.
Finally some of the tables must be better presented. Figure 1 must be presented with the number of patients excluded and the number of excluded patients due to each exclusion criteria would make it a better figure. Some patients are lost during the follow-up which is natural in a follow-up study. But it will be a better paper if the reader can see why the patients are lost-to-follow-up. And some of the headings in the tables must be in English and not in Polish.
Author Response
Dear Reviewer #2,
Thank you for reviewing our article titled ‘The prevalence of back pain in patients operated on due to colorectal cancer depending on the type of surgical procedure performed ‘ We deeply appreciate your opinion as well as constructive comments that contributed to more profound consideration of issues addressed in our publication. The comments in your review will guide us in our future work.
In response to those commentaries we have clarified the following points:
- We have described the process of interviewing patients in Material and Method and Discussion section
The responses at the latter two time points were collected by means of the CATI- Computer Assisted Telephone Interviewing method with the patients being contacted by telephone due to the ongoing coronavirus pandemic. Patients, by signing informed consent, knew at what point in the procedure telephone contact would occur. The persons interviewing the patients were the 4 authors of the manuscript. In order to avoid bias, the interviewers did not have data on the type of surgical procedure the patient had undergone.
- We have addend the information in Fig 1 why the patient were lost-to-follow-up. We corrected the headings in tables and figures and improved the visibility of the figures.
Thanks to the comments received, we were able to refine the publication in terms of the content. All errors mentioned in the review were corrected in the final version of the publication.
Kind regards,
the Authors

Round 2
Reviewer 1 Report
The authors have appropriately replied to the reviewer's comments in a point-to-point manner and have revised the manuscript accordingly.
Reviewer 2 Report
Dear authors.
Thanks for the revised mansuscript and the corrections you have made. Looking forward to see it published.